# Canonical Capsules:
# Self-Supervised Capsules in Canonical Pose

**Weiwei Sun**[1,4,*]    **Andrea Tagliasacchi**[2,3,*]    **Boyang Deng**[3]    **Sara Sabour**[2,3]

**Soroosh Yazdani**[3]    **Geoffrey Hinton**[2,3]    **Kwang Moo Yi**[1,4]

[1]University of British Columbia,   [2]University of Toronto,
[3]Google Research,   [4]University of Victoria,   [*]equal contributions

https://canonical-capsules.github.io

## Abstract

We propose a self-supervised capsule architecture for 3D point clouds. We compute capsule decompositions of objects through permutation-equivariant attention, and self-supervise the process by training with pairs of randomly rotated objects. Our key idea is to aggregate the attention masks into semantic keypoints, and use these to supervise a decomposition that satisfies the capsule invariance/equivariance properties. This not only enables the training of a semantically consistent decomposition, but also allows us to learn a canonicalization operation that enables object-centric reasoning. To train our neural network we require neither classification labels nor manually-aligned training datasets. Yet, by learning an object-centric representation in a self-supervised manner, our method outperforms the state-of-the-art on 3D point cloud reconstruction, canonicalization, and unsupervised classification.

## 1   Introduction

Understanding objects is one of the core problems of computer vision [32, 14, 38]. While this task has traditionally relied on large annotated datasets [42, 22], unsupervised approaches that utilize self-supervision [5] have emerged to remove the need for labels. Recently, researchers have attempted to extend these methods to work on 3D point clouds [59], but the field of unsupervised 3D learning remains relatively uncharted. Conversely, researchers have been extensively investigating 3D deep representations for shape auto-encoding[1] [61, 19, 33, 16], making one wonder whether these discoveries can now benefit from unsupervised learning for tasks *other* than auto-encoding.

Importantly, these recent methods for 3D deep representation learning are not entirely unsupervised. Whether using point clouds [61], meshes [19], or implicits [33], they owe much of their success to the bias within the dataset that was used for training. Specifically, all 3D models in the popular ShapeNet [3] dataset are "object-centric" – they are pre-canonicalized to a unit bounding box, and, even more importantly, with an orientation that synchronizes object semantics to Euclidean frame axes (e.g. airplane cockpit is always along $+y$, car wheels always touch $z = 0$). Differentiable 3D decoders are heavily affected by the consistent alignment of their output with an Euclidean frame [8, 16] as local-to-global transformations *cannot* be easily learnt by fully connected layers. As we will show in Section 4.2, these methods fail in the absence of pre-alignment, even when data

---

[1]Auto-encoding is also at times referred to as "reconstruction" or "shape-space" learning.

35th Conference on Neural Information Processing Systems (NeurIPS 2021), virtual.

augmentation is used. A concurrent work [45] also recognizes this problem and proposes a separate learnt canonicalizer, which is shown helpful in downstream classification tasks.

In this work, we leverage the modeling paradigm of capsule networks [23]. In capsule networks, a scene is perceived via its decomposition into *part* hierarchies, and each part is represented with a (pose, descriptor) pair: ① The capsule *pose* specifies the frame of reference of a part, and hence should be transformation equivariant; ② The capsule *descriptor* specifies the appearance of a part, and hence should be transformation invariant. Thus, one does not have to worry how the data is oriented or translated, as these changes can be encoded within the capsule representation.

We introduce *Canonical **Capsules***, a novel capsule architecture to compute a K-part decomposition of a point cloud. We train our network by feeding pairs of a randomly rotated/translated copies of the same shape (i.e. siamese training) hence removing the requirement of pre-aligned training datasets. We then decompose the point cloud by assigning each point into one of the K parts via attention, which we aggregate into K keypoints. Equivariance is then enforced by requiring the two keypoint sets to only differ by the known (relative) transformation (i.e. a form of self-supervision). For invariance, we simply ask that the descriptors of each keypoint of the two instances match.

With *Canonical Capsules*, we exploit our decomposition to recover a canonical frame that allows unsupervised "object-centric" learning of 3D deep representations *without* requiring a semantically aligned dataset. We achieve this task by regressing *canonical* capsule poses from capsule descriptors via a deep network, and computing a canonicalizing transformation by solving a shape-matching problem [44]. This not only allows more effective shape auto-encoding, but our experiments confirm this results in a latent representation that is more effective in unsupervised classification tasks. Note that, like our decomposition, our canonicalizing transformations are also learnt in a self-supervised fashion, by only training on randomly transformed point clouds.

**Contributions**. In summary, in this paper we:

- propose an architecture for 3D self-supervised learning based on capsule decomposition;
- enable object-centric unsupervised learning by introducing a learned canonical frame of reference;
- achieve state-of-the-art performance 3D point cloud auto-encoding/reconstruction, canonicalization, and unsupervised classification.

## 2   Related works

Convolutional Neural Networks lack equivariance to rigid transformations, despite their pivotal role in describing the structure of the 3D scene behind a 2D image. One promising approach to overcome this shortcoming is to add equivariance under a group action in each layer [49, 9]. In our work, we remove the need for a global $\mathbf{SE}(3)$-equivariant network by canonicalizing the input.

**Capsule Networks**. Capsule Networks [23] have been proposed to overcome this issue towards a relational and hierarchical understanding of natural images. Of particular relevance to our work, are methods that apply capsule networks to 3D input data [64, 65, 46], but note these methods are not unsupervised, as they either rely on classification supervision [65], or on datasets that present a significant inductive bias in the form of pre-alignment [64]. In this paper, we take inspiration from the recent Stacked Capsule Auto-Encoders [30], which shows how capsule-style reasoning can be effective as long as the *primary* capsules can be trained in a self-supervised fashion (i.e. via reconstruction losses). The natural question, which we answer in this paper, is *"How can we engineer networks that generate 3D primary capsules in an unsupervised fashion?"*

**Deep 3D representations**. Reconstructing 3D objects requires effective inductive biases about 3D vision *and* 3D geometry. When the input is images, the core challenge is how to encode *3D projective geometry* concepts into the model. This can be achieved by explicitly modeling multi-view geometry [28], by attempting to learn it [13], or by hybrid solutions [60]. But even when input is 3D, there are still significant challenges. It is still not clear which is the 3D *representation* that is most amenable to deep learning. Researchers proposed the use of meshes [53, 31], voxels [54, 55], surface patches [19, 12, 10], and implicit functions [35, 33, 7]. Unfortunately, the importance of geometric structures (i.e. *part-to-whole* relationships) is often overlooked. Recent works have tried to close this gap by using part decomposition consisting of oriented boxes [50], ellipsoids [17, 16], convex polytopes [11], and grids [2]. However, as previously discussed, most of these still heavily rely on a

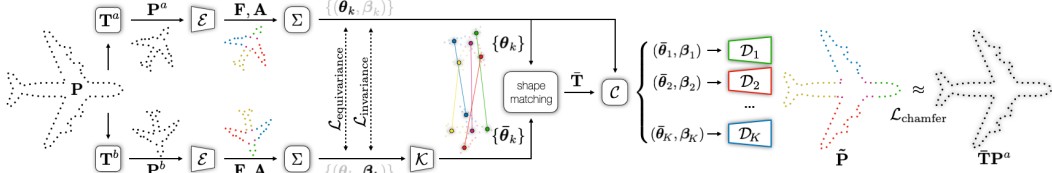

Figure 1: **Framework** – We learn a capsule encoder for 3D point clouds by relating the decomposition result of two random rigid transformations $\mathbf{T}^a$ and $\mathbf{T}^b$, of a given point cloud, *i.e.*, a Siamese training setup. We learn the parameters of an encoder $\mathcal{E}$, a per-capsule decoder $\mathcal{D}_k$, as well as a network that represents a learnt canonical frame $\mathcal{K}$. For illustrative purposes, we shade-out the outputs that do not flow forward, and with $\Sigma$ summarize the aggregations in (2).

pre-aligned training dataset; our paper attempts to bridge this gap, allowing learning of *structured* 3D representations *without* requiring pre-aligned data.

**Canonicalization**. One way to circumvent the requirement of pre-aligned datasets is to rely on methods capable of registering a point cloud into a canonical frame. The recently proposed CaSPR [39] fulfills this premise, but requires ground-truth canonical point clouds in the form of normalized object coordinate spaces [52] for supervision. Similarly, [20] regresses each view's pose relative to the canonical pose, but still requires weak annotations in the form of multiple partial views. C3DPO [34] learns a canonical 3D frame based on 2D input keypoints. In contrast to these methods, our solution is completely self-supervised. The concurrent Compass [45] also learns to canonicalize in a self-supervised fashion, but as the process is not end-to-end, this results in a worse performance than ours, as it will be shown in Section 4.3.

**Registration**. Besides canonicalization, many *pairwise* registration techniques based on deep learning have been proposed (e.g. [56, 63]), even using semantic keypoints and symmetry to perform the task [15]. These methods typically register a *pair* of instances from the same class, but lack the ability to *jointly* and consistently register all instances to a shared canonical frame.

## 3 Method

Our network trains on unaligned point clouds as illustrated in Figure 1: we train a network that *decomposes* point clouds into parts, and enforce invariance/equivariance through a Siamese training setup [48]. We then *canonicalize* the point cloud to a learnt frame of reference, and perform *auto-encoding* in this coordinate space. The losses employed to train $\mathcal{E}$, $\mathcal{K}$, and $\mathcal{D}$, will be covered in Section 3.1, while the details of their architecture are in Section 3.2.

**Decomposition**. In more detail, given a point cloud $\mathbf{P} \in \mathbb{R}^{P \times D}$ of $P$ points in $D$ dimensions (in our case $D{=}3$), we perturb it with two random transformations $\mathbf{T}^a, \mathbf{T}^b \in \mathbf{SE}(D)$ to produce point clouds $\mathbf{P}^a, \mathbf{P}^b$. We then use a shared permutation-equivariant capsule encoder $\mathcal{E}$ to compute a $K$-fold attention map $\mathbf{A} \in \mathbb{R}^{P \times K}$ for $K$ capsules, as well as per-point feature map $\mathbf{F} \in \mathbb{R}^{P \times C}$ with $C$ channels:

$$\mathbf{A}, \mathbf{F} = \mathcal{E}(\mathbf{P}) , \tag{1}$$

where we drop the superscript indexing the Siamese branch for simplicity. From these attention masks, we then compute, for the $k$–th capsule its pose $\boldsymbol{\theta}_k \in \mathbb{R}^3$ parameterized by its location in 3D space, and the corresponding capsule descriptor $\boldsymbol{\beta}_k \in \mathbb{R}^C$:

$$\boldsymbol{\theta}_k = \frac{\sum_p \mathbf{A}_{p,k} \mathbf{P}_p}{\sum_p \mathbf{A}_{p,k}} , \qquad \boldsymbol{\beta}_k = \frac{\sum_p \mathbf{A}_{p,k} \mathbf{F}_p}{\sum_p \mathbf{A}_{p,k}} . \tag{2}$$

Hence, as long as $\mathcal{E}$ is invariant w.r.t. rigid transformations of $\mathbf{P}$, the pose $\boldsymbol{\theta}_k$ will be transformation equivariant, and the descriptor $\boldsymbol{\beta}_k$ will be transformation invariant. Note that this simplifies the design (and training) of the encoder $\mathcal{E}$, which only needs to be invariant, rather than equivariant [49, 46].

**Canonicalization**. Simply enforcing invariance and equivariance with the above framework is not enough to learn 3D representations that are object-centric, as we lack an (unsupervised) mechanism to bring information into a shared "object-centric" reference frame. Furthermore, the "right" canonical

frame is nothing but a convention, thus we need a mechanism that allows the network to make a choice – a choice, however, that must then be consistent across all objects. For example, a learnt canonical frame where the cockpit of airplanes is consistently positioned along $+z$ is *just as good* as a canonical frame where it is positioned along the $+y$ axis. To address this, we propose to link the capsule descriptors to the capsule poses in canonical space, that is, we ask that objects with similar appearance to be located in similar Euclidean neighborhoods in canonical space. We achieve this by regressing canonical capsules poses (i.e. canonical keypoints) $\bar{\boldsymbol{\theta}} \in \mathbb{R}^{K \times 3}$ using the descriptors $\boldsymbol{\beta} \in \mathbb{R}^{K \times C}$ via a fully connected deep network $\mathcal{K}$:

$$\bar{\boldsymbol{\theta}} = \mathcal{K}(\boldsymbol{\beta}) \tag{3}$$

Because fully connected layers are biased towards learning low-frequency representations [27], this regressor also acts as a regularizer that enforces semantic locality.

**Auto-encoding**. Finally, in the learnt *canonical* frame of reference, to train the capsule descriptors via auto-encoding, we reconstruct the point clouds with per-capsule decoders $\mathcal{D}_k$:

$$\tilde{\mathbf{P}} = \cup_k \left\{ \mathcal{D}_k(\bar{\mathbf{R}}\boldsymbol{\theta}_k + \bar{\mathbf{t}}, \boldsymbol{\beta}_k) \right\} , \tag{4}$$

where $\cup$ denotes the union operator. The canonicalizing transformation $\bar{\mathbf{T}} = (\bar{\mathbf{R}}, \bar{\mathbf{t}})$ can be readily computed by solving a shape-matching problem [44], thanks to the property that our capsule poses and regressed keypoints are in *one-to-one correspondence*:

$$\bar{\mathbf{R}}, \bar{\mathbf{t}} = \arg\min_{\mathbf{R}, \mathbf{t}} \frac{1}{K} \sum_k \|(\mathbf{R}\boldsymbol{\theta}_k + \mathbf{t}) - \bar{\boldsymbol{\theta}}_k\|_2^2 . \tag{5}$$

While the reconstruction in (4) is in canonical frame, note it is trivial to transform the point cloud back to the original coordinate system after reconstruction, as the transformation $\bar{\mathbf{T}}^{-1}$ is available.

## 3.1 Losses

As common in unsupervised methods, our framework relies on a number of losses that control the different characteristics we seek to obtain in our representation. Note none of these losses require ground truth labels. We organize the losses according to the portion of the network they supervise: *decomposition*, *canonicalization*, and *reconstruction*.

**Decomposition**. While a transformation invariant encoder architecture should be sufficient to achieve equivariance/invariance, this does not prevent the encoder from producing trivial solutions/decompositions once trained. As capsule poses should be *transformation equivariant*, the poses of the two rotation augmentations $\boldsymbol{\theta}_k^a$ and $\boldsymbol{\theta}_k^b$ should only differ by the (known) relative transformation:

$$\mathcal{L}_{\text{equivariance}} = \frac{1}{K} \sum_k \|\boldsymbol{\theta}_k^a - (\mathbf{T}^a)(\mathbf{T}^b)^{-1}\boldsymbol{\theta}_k^b\|_2^2 . \tag{6}$$

Conversely, capsule descriptors should be *transformation invariant*, and as the two input points clouds are of the *same* object, the corresponding capsule descriptors $\boldsymbol{\beta}$ should be identical:

$$\mathcal{L}_{\text{invariance}} = \frac{1}{K} \sum_k \|\boldsymbol{\beta}_k^a - \boldsymbol{\beta}_k^b\|_2^2 . \tag{7}$$

We further regularize the capsule decomposition to ensure each of the $K$ heads roughly represent the same "amount" of the input point cloud, hence preventing degenerate (zero attention) capsules. This is achieved by penalizing the attention *variance*:

$$\mathcal{L}_{\text{equilibrium}} = \frac{1}{K} \sum_k \|a_k - \tfrac{1}{K}\Sigma_k a_k\|_2^2 , \tag{8}$$

where $a_k = \Sigma_p(\mathbf{A}_{p,k})$ denotes the total attention exerted by the k-th head on the point cloud.

Finally, to facilitate the training process, we ask for capsules to learn a localized representation of geometry. We express the spatial extent of a capsule by computing first-order moments of the represented points with respect to the capsule pose $\boldsymbol{\theta}_k$:

$$\mathcal{L}_{\text{localization}} = \frac{1}{K} \sum_k \tfrac{1}{a_k} \sum_p \mathbf{A}_{p,k} \|\boldsymbol{\theta}_k - \mathbf{P}_p\|_2^2 . \tag{9}$$

**Canonicalization**. To train our canonicalizer $\mathcal{K}$, we relate the predicted capsule poses to regressed canonical capsule poses via the *optimal* rigid transformation from (5):

$$\mathcal{L}_{\text{canonical}} = \frac{1}{K} \sum_k \|(\bar{\mathbf{R}}\boldsymbol{\theta}_k + \bar{\mathbf{t}}) - \bar{\boldsymbol{\theta}}_k\|_2^2 \; . \tag{10}$$

Recall that $\bar{\mathbf{R}}$ and $\bar{\mathbf{T}}$ are obtained through a differentiable process. Thus, this loss is forcing the aggregated pose $\boldsymbol{\theta}_k$ to agree with the one that goes through the regression path, $\bar{\boldsymbol{\theta}}_k$. Now, since $\bar{\boldsymbol{\theta}}_k$ is regressed solely from the set of capsule descriptors, similar shapes will result in similar canonical keypoints, and the coordinate system of $\bar{\boldsymbol{\theta}}_k$ is one that employs Euclidean space to encode semantics.

**Reconstruction**. To learn canonical capsule descriptors in an unsupervised fashion, we rely on an auto-encoding task. We train the decoders $\{\mathcal{D}_k\}$ by minimizing the *Chamfer Distance* (CD) between the (canonicalized) input point cloud and the reconstructed one, as in [61, 19]:

$$\mathcal{L}_{\text{recon}} = \text{CD}\left(\bar{\mathbf{R}}\mathbf{P} + \bar{\mathbf{t}}, \; \tilde{\mathbf{P}}\right) \; . \tag{11}$$

## 3.2 Network Architectures

We briefly summarize our implementation details, including the network architecture; for further details, please refer to the `supplementary material`.

**Encoder – $\mathcal{E}$**. Our architecture is based on the one suggested in [47]: a pointnet-like architecture with residual connections and attentive context normalization. We utilize Batch Normalization [26] instead of the Group Normalization [58], which trained faster in our experiments. We further extend their method to have *multiple* attention maps, where each attention map corresponds to a capsule.

**Decoder – $\mathcal{D}$**. The decoder from (4) operates on a per-capsule basis. Our decoder architecture is similar to AtlasNetV2 [12] (with trainable grids). The difference is that we translate the per-capsule decoded point cloud by the corresponding capsule pose.

**Regressor – $\mathcal{K}$**. We concatenate the descriptors and apply a series of fully connected layers with ReLU activation to regress the $P$ capsule locations. At the output layer we use a linear activation and subtract the mean of the outputs to make our regressed locations zero-centered in the canonical frame.

**Canonicalizing the descriptors**. As our descriptors are only *approximately* rotation invariant (via $\mathcal{L}_{\text{invariance}}$), we found it useful to re-extract the capsule descriptors $\boldsymbol{\beta}_k$ after canonicalization. Specifically, we compute $\bar{\mathbf{F}}$ with the same encoder setup, but with $\bar{\mathbf{P}}=\bar{\mathbf{R}}\mathbf{P}+\bar{\mathbf{T}}$ instead of $\mathbf{P}$ and use it to compute $\bar{\boldsymbol{\beta}}_k$; we validate this empirically in the `supplementary material`.

# 4 Results

We first discuss the experimental setup, and then validate our method on a *variety* of tasks: auto-encoding, canonicalization, and unsupervised classification. While the task differs, our learning process remains the *same*: we learn capsules by reconstructing objects in a learnt canonical frame. We also provide an ablation study, which is expanded in detail in the `supplementary material`, where we provide additional qualitative results.

## 4.1 Experimental setup

To evaluate our method, we rely on the ShapeNet (Core) dataset [3][2]. We follow the category choices from AtlasNetV2 [12], using the airplane and chair classes for *single-category* experiments, while for *multi-category* experiments we use all 13 classes: airplane, bench, cabinet, car, chair, monitor, lamp, speaker, firearm, couch, table, cellphone, and watercraft. To make our results most compatible with those reported in the literature, we also use the same splits as in AtlasNetV2 [12]: 31747 shapes in the train, and 7943 shapes in the test set. Unless otherwise noted, we randomly sample 1024 points from the object surface for each shape to create our 3D point clouds.

**De-canonicalizing the dataset**. As discussed in the introduction, ShapeNet (Core) contains substantial inductive bias in the form of consistent semantic alignment. To remove this bias, we create random

---

[2]Available to researchers for non-commercial research and educational use.

**SE**(3) transformations, and apply them to each point cloud. We first generate uniformly sampled random rotations, and add uniformly sampled random translations within the range $[-0.2, 0.2]$, where the bounding volume of the shape ranges in $[-1, +1]$. Note the relatively limited translation range is chosen to give state-of-the-art methods a *chance* to compete with our solution. We then use the relative transformation between the point clouds extracted from this ground-truth transformation to evaluate our methods. We refer to this unaligned version of the ShapeNet Core dataset as the *unaligned* setup, and using the vanilla ShapeNet Core dataset as the *aligned* setup. For the *aligned* setup, as there is no need for equivariance adaptation, we simply train our method without the random transformations, and so $\mathcal{L}_{\text{equivariance}}$ and $\mathcal{L}_{\text{invariance}}$ are not used. This setup is to simply demonstrate how Canonical Capsules would perform in the presence of a dataset bias.

We emphasize here that a proper generation of random rotation is important. While some existing works have generated them by uniformly sampling the degrees of freedom of an Euler-angle representation, this is known to be an incorrect way to sample random rotations [41], leading to biases in the generated dataset; see the `supplementary material`.

**Implementation details**. For all our experiments we use the Adam optimizer [29] with an initial learning rate of $0.001$ and decay rate of $0.1$. We train for 325 epochs for the *aligned* setup to match the AtlasNetV2 [12] original setup. For the *unaligned* setting, as the problem is harder, we train for a longer number of 450 epochs. We use a batch size of 16. The training rate is $\sim$2.5 iters/sec. We train each model on a single NVidia V100 GPU. Unless stated otherwise, we use $k$=10 capsules and capsule descriptors of dimension $C$=128. We train three models with our method: two that are single-category (*i.e.*, for airplane and chairs), and one that is multi-category (*i.e.*, all 13 classes). To set the weights for each loss term, we rely on the reconstruction performance (CD) in the training set. We set weights to be one for all terms except for $\mathcal{L}_{\text{equivariance}}$ (5) and $\mathcal{L}_{\text{equilibrium}}$ ($10^{-3}$). In the aligned case, because $\mathcal{L}_{\text{equivariance}}$ and $\mathcal{L}_{\text{invariance}}$ are not needed (always zero), we reduce the weights for the other decomposition losses by $10^3$; $\mathcal{L}_{\text{localization}}$ to $10^{-3}$ and $\mathcal{L}_{\text{equilibrium}}$ to $10^{-6}$.

### 4.2 Auto-encoding – Figure 2 and Table 1

We evaluate the performance of our method for the task that was used to train the network – reconstruction / auto-encoding – against three baselines (trained in both single-category and multi-category variants): ① 3D-PointCapsNet [64], an auto-encoder for 3D point clouds that utilize a capsule architecture; ② AtlasNetV2 [12], a state-of-the-art auto-encoder which utilizes a multi-head patch-based decoder; ③ AtlasNetV2 [12] with a spatial-transformer network (STN) aligner from PointNet [36], a baseline with canonicalization. We do not compare against [46], as unfortunately source code is not publicly available.

Table 1: **Auto-encoding / quantitative –** Performance in terms of Chamfer distance with 1024 points per point cloud – metric is multiplied by $10^3$ as in [12].

|  | Method | Airplane | Chair | Multi |
|---|---|---|---|---|
| Aligned | 3D-PointCapsNet [64] | 1.94 | 3.30 | 2.49 |
| | AtlasNetV2 [12] | 1.28 | 2.36 | 2.14 |
| | Our method | **0.96** | **1.99** | **1.76** |
| Unaligned | 3D-PointCapsNet [64] | 5.58 | 7.57 | 4.66 |
| | AtlasNetV2 [12] | 2.80 | 3.98 | 3.08 |
| | AtlasNetV2 [12] w/ STN | 1.90 | 2.99 | 2.60 |
| | Our method | **1.11** | **2.58** | **2.22** |

**Quantitative analysis – Table 1**. We achieve state-of-the-art performance in both the *aligned* and *unaligned* settings. The wider margin in the *unaligned* setup indicates tackling this more challenging scenario damages the performance of AtlasNetV2 [12] and 3D-PointCapsNet [64] much more than our method[3]. We also include a variant of AtlasNetV2 for which a STN (Spatial Transformer Network) is used to pre-align the point clouds [36], demonstrating how the simplest form of pre-aligner/canonicalizer is not sufficient.

**Qualitative analysis – Figure 2**. We illustrate our decomposition-based reconstruction of 3D point clouds, as well as the reconstructions of 3D-PointCapsNet [64] and AtlasNetV2 [12]. As shown, even in the *unaligned* setup, our method is able to provide *semantically consistent* capsule decompositions – *e.g.* the wings of the airplane have consistent colours, and when aligned in the canonical frame, the different airplane instances are well-aligned. Compared to AtlasNetV2 [12] and 3D-PointCapsNet [64], the reconstruction quality is also visibly improved: we better preserve details along the engines of the airplane, or the thin structures of the bench; note also that the decompositions

---

[3]Results in this table differ slightly from what is reported in the original papers as we use 1024 points to speed-up experiments throughout our paper. However, in the `supplementary material` the same trend holds regardless of the number of points, and match with what is reported in the original papers with 2500 points.

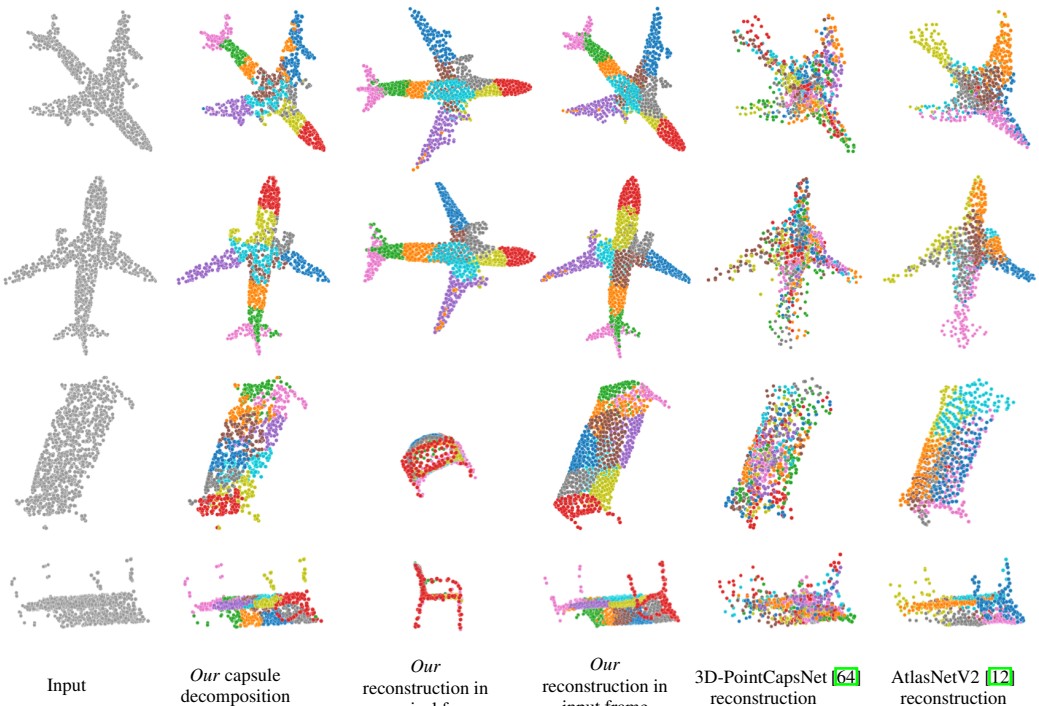

| Input | *Our* capsule decomposition | *Our* reconstruction in canonical frame | *Our* reconstruction in input frame | 3D-PointCapsNet [64] reconstruction | AtlasNetV2 [12] reconstruction |

Figure 2: **Auto-encoding / qualitative –** Example decomposition and reconstruction results setup using Canonical Capsules on several *unaligned* point cloud instances from the test set. We color each Canonical Capsule with a unique colour, and similarly color "patches" from the reconstruction heads of 3D-PointCapsNet [64] and AtlasNetV2 [12]. Canonical Capsules provide semantically consistent decomposition that is aligned in canonical frame, leading to improved reconstruction quality.

are semantically consistent in our examples. Results are better appreciated in our `supplementary material`, where we visualize the performance as we continuously traverse $\mathbf{SE}(3)$.

### 4.3 Canonicalization – Table 2

We compare against three baselines: ① Deep Closest Points [56], a deep learning-based *pairwise* point cloud registration method; ② DeepGMR [63] a state-of-the-art *pairwise* registration method that decomposes clouds into Gaussian mixtures and utilizes Rigorously Rotation-Invariant (RRI) features [4]; ③ Compass [45] a concurrent work on learnt alignment/canonicalization. For all compared methods we use the official implementation. For DeepGMR we use both RRI and the typical XYZ coordinates as input. We also try our method with the RRI features, following DeepGMR's training protocol and train for 100 epochs.

**Metrics**. To evaluate the canonicalization performance, we look into the stability of the canonicalization – the shakiness shown in the videos in our `supplementary material`– represented as the mean standard deviation of the rotations (mStd):

$$\text{mStd} = \frac{1}{n} \sum_{i=1}^{n} \sqrt{\frac{1}{m} \sum_{j=1}^{m} (\angle(\mathbf{R}^{ij}, \mathbf{R}^{i}_{mean}))^2} , \tag{12}$$

where $\angle$ is the angular distance between two rotation matrices [18, 62, 63], $\mathbf{R}^{ij}$ is the rotation matrix of the $j$-th instance of the $i$-th object in canonical frame, and $\mathbf{R}^{i}_{mean}$ is the mean rotation [21] of the $i$-th object. Note that with mStd we measure the stability of canonicalization with respect to rotations to accommodate for methods that do not deal with translation [45]. To allow for comparisons with pairwise registration methods, we also measure performance in terms of the RMSE metric [63, 66].

**Quantitative analysis – Table 2.** Compared to Compass [45], our method provides improved stability in canonicalization. This also provides an advantage in pairwise registration, delivering state-of-the-art results when XYZ-coordinates are used. Note that while *pairwise* methods can align

Table 2: **Canonicalization** – Quantitative evaluation for canonicalization, where we highlight significantly better performance than the concurrent work Compass [45]. While pairwise registration is not our main objective, the *global* alignment frame created by our method still allows for effective registration (on par, or better) than the state-of-the-art.

| | Method | Canonicalization (mStd) ↓ | | | Pairwise registration (RMSE) ↓ | | |
| --- | --- | --- | --- | --- | --- | --- | --- |
| | | Airplane | Chair | Multi | Airplane | Chair | Multi |
| XYZ-coord. | Deep Closest Points [56] | – | – | – | 0.318 | 0.160 | 0.131 |
| | DeepGMR [63] | – | – | – | 0.079 | 0.082 | 0.077 |
| | Compass [45] | 19.105 | 19.508 | 51.811 | 0.412 | 0.482 | 0.515 |
| | Our method | **8.278** | **5.730** | **21.210** | **0.022** | **0.027** | **0.074** |
| RRI | DeepGMR [63] | – | – | – | **0.0001** | **0.0001** | **0.0001** |
| | Our method | *(unstable)* | *(unstable)* | *(unstable)* | 0.0006 | 0.0009 | 0.0016 |

two sets of given point clouds, creating a canonical frame that simultaneously registers *all* point clouds is a non-trivial extension to the problem.

When RRI is used as input, our method is on par with DeepGMR [63], up to a level where registration is near perfect – alignment differences when errors are in the $10^{-4}$ ballpark are indiscernible. We note that the performance of Deep Closest Points [56] is not as good as reported in the original paper, as we uniformly draw rotations from $\mathbf{SO}(3)$. When a sub-portion of $\mathbf{SO}(3)$ is used, *e.g.* a quarter of what we are using, DCP performs relatively well (0.008 in the multi-class experiment). While curriculum learning could be used to enhance the performance of DCP, our technique does not need to rely on these more complex training techniques.

We further note that, while RRI delivers good registration performance, using RRI features cause the learnt canonicalization to fail – training becomes *unstable*. This hints that RRI features may be throwing away too much information to achieve transformation invariance. Our method using raw XYZ coordinates as input, on the other hand, provides comparable registration performance, and is able to do significantly more than just registration (*i.e.* classification, reconstruction).

## 4.4  Unsupervised classification – Table 3

Beyond reconstruction and canonicalization, we evaluate the usefulness of our method via a classification task that is not related in *any way* to the losses used for training. We compute the features from the auto-encoding methods from Section 4.2 against those from our method (where we build features by combining pose with descriptors) to perform 13-way classification with two different techniques:

- We train a *supervised* linear Support Vector Machine (SVM) on the extracted features [1, Ch. 7];
- We perform *unsupervised* K-Means clustering [1, Ch. 9] and then label each cluster via bipartite matching with the actual labels through the Hungarian algorithm.

Note the former provides an upper bound for unsupervised classification, while better performance on the latter implies that the learnt features are able to separate the classes into clusters that are compact (in an Euclidean sense).

Table 3: **Classification** – Top-1 accuracy(%)

| | Method | SVM | K-Means |
| --- | --- | --- | --- |
| Aligned | 3D-PointCapsNet [64] | 93.81 | 65.87 |
| | AtlasNetV2 [12] | 94.07 | 61.66 |
| | Our method | **94.21** | **69.82** |
| Unaligned | 3D-PointCapsNet [64] | 71.13 | 14.59 |
| | AtlasNetV2 [12] | 64.85 | 17.12 |
| | AtlasNetV2 [12] w/ STN | 78.55 | 20.03 |
| | Our method | **87.33** | **43.04** |

**Analysis of results – SVM**. Note how our method provides best results in all cases, and when the dataset is not unaligned the difference is significant. This shows that, while 3D-PointCapsNet and AtlasNetV2 (with and without STN) are able to somewhat auto-encode point clouds in the *unaligned* setup, what they learn does not translate well to classification. However, the features learned with Canonical Capsules are more related to the semantics of the object, which helps classification.

**Analysis of results – K-Means**. The performance gap becomes wider when K-Means is used – even in the *aligned* case. This could mean that the features extracted by Canonical Capsules are better suited for other unsupervised tasks, having a feature space that is close to being Euclidean in terms of semantics. The difference is striking in the *unaligned* setup. We argue that these results emphasize the importance of the capsule framework – jointly learning the invariances and equivariances in the data – is cardinal to unsupervised learning [25, 24].

Table 4: **Effect of losses –** Reconstruction performance, and canonicalization performance when loss terms are removed.

| | Full | $\neg\mathcal{L}_{\text{invar}}$ | $\neg\mathcal{L}_{\text{equiv}}$ | $\neg\mathcal{L}_{\text{canonical}}$ | $\neg\mathcal{L}_{\text{localization}}$ | $\neg\mathcal{L}_{\text{equilibrium}}$ |
|---|---|---|---|---|---|---|
| CD | **1.11** | 1.12 | 1.16 | 1.12 | 1.44 | 1.60 |
| Std | **8.278** | 9.983 | 110.174 | 8.421 | 113.204 | 92.970 |

Table 5: **Backbone –** Auto-encoding performance (Chamfer distance) when we use various permutation-invariant backbones.

| | PointNet | PointNet++ | DGCNN | ACNe |
|---|---|---|---|---|
| CD | 1.28 | 1.34 | 1.21 | **1.11** |

### 4.5 Ablation study

To make the computational cost manageable, we perform all ablations with the *airplane* category (the category with most instances), and in the *unaligned* setup (unless otherwise noted). Please also see `supplementary material` for more ablation studies.

**Losses – Table 4**. We analyze the importance of each loss term, with the exception of $\mathcal{L}_{\text{recon}}$ which is necessary for training. All losses beneficially contribute to reconstruction performance, but note how $\mathcal{L}_{\text{equiv}}$, $\mathcal{L}_{\text{localization}}$ and $\mathcal{L}_{\text{equilibrium}}$ affect it to a larger degree. By considering our canonicalization metric, we can motivate this outcome by observing that the method fails to perform canonicalization when these losses are not employed (i.e. training collapses).

**Encoder architecture – Table 5**. Our method can be used with *any* backbone, as our main contribution lies in the self-supervised canonicalization architecture. For completeness, we explore variants of our method using different back-bones: PointNet [36], PointNet++ [37], DGCNN [57], and ACNe [47]. Among these, the ACNe backbone performs best; note that all the variants in Table 5, *regardless* of backbone choice, significantly outperforms all other methods reported in Table 1.

## 5 Conclusions

In this paper, we provide a self-supervised framework to train *primary* capsule decompositions for 3D point clouds. We rely on a Siamese setup that allows self-supervision and auto-encoding in canonical space, circumventing the customary need to train on pre-aligned datasets. Despite being trained in a self-supervised fashion, our representation achieves state-of-the-art performance across auto-encoding, canonicalization and classification tasks. These results are made possible by allowing the network to learn a canonical frame of reference. We interpret this result as giving our neural networks a mechanism to construct a "mental picture" of a given 3D object – so that downstream tasks are executed within an object-centric coordinate frame.

**Future work**. As many objects have natural symmetries that we do not consider at the moment [12], providing our canonicalizer a way to encode such a prior is likely to further improve the representation. We perform decomposition at a single level, and it would be interesting to investigate how to effectively engineer multi-level decompositions [51]; one way could be to over-decompose the input in a redundant fashion (with large $K$), and use a downstream layers that "selects" the decomposition heads to be used [6]. We would also like to extend our results to more "in-the-wild" 3D computer vision and understand whether learning object-centric representations is possible when *incomplete* (*i.e.*, single view [35] or occluded) data is given in input, when an entire scene with potentially *multiple* objects is given [43], or where our measurement of the 3D world is a single 2D image [48], or by exploiting the persistence of objects in video [40].

**Broader impact**. While our work is exclusively on 3D shapes, and thus not immediately subject to any societal concerns, it enhances how Artificial Intelligence (AI) can understand and model 3D geometry. Thus, similar to how image recognition could be misused, one should be careful when extending the use of our method. In addition, while not subject to how the data itself is aligned, the learnt canonical frame of our method is still data-driven, thus subject to any data collection biases that may exist – canonicalization will favour shapes that appear more often. This should also be taken into account with care to prevent any biased decisions when utilizing our method within a decision making AI platform.

## Acknowledgements

This work was supported by the Natural Sciences and Engineering Research Council of Canada (NSERC) Discovery Grant, NSERC Collaborative Research and Development Grant, Google, Compute Canada, and Advanced Research Computing at the University of British Columbia.

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
