# OpenReview forum: "Canonical Capsules: Self-Supervised Capsules in Canonical Pose"
_NeurIPS.cc/2021/Conference — NeurIPS 2021 Poster_

### Official Review · Reviewer_jD2y · 2021-07-14

**Rating:** 7
**Confidence:** 4

**Summary:**

The paper proposed a self-supervised capsule based architecture for 3D point cloud. The main goal is to learn a  canonicalization operation that allows unbiased object-centric reasoning.  The idea is that an object is decomposed to K keypoints using canonical capsules, the whole system is trained in an unsupervised fashion based on pairs of randomly rotated/translated  of the same object. The capsule pose equivariance is enforced by requiring the equivariance of two keypoint sets based on the known relative transformation of the pair images and the capsule descriptor invariance is enforced by matching the pair keypoints. The proposed framework have been evaluated in 3D autoencoding and reconstruction, unsupervised classification and canonicalization on ShapeNet dataset.

**Ethical Concerns:**

I don't see any ethical issue with the paper.

**Limitations And Societal Impact:**

The authors addressed the limitation and societal impact adequately in the future work and broader impact subsections.

**Main Review:**

Pros:
- The paper is generally well written and easy to follow.
- The unsupervised learning framework for 3D canonicalization is very novel and seems effective in classification, reconstruction and canonicalization tasks.
- The proposed framework outperforms the baseline in all 3 tasks.

Cons/ questions/ clarifications:
- Would you please clarify the encoder architecture? Does it only have one capsule layer with K capsule types as its last layer?
- What is the effect of K on the performance and have you considered adding capsule layers as intermediate layers.
- While the limited translation range is in favor of baseline methods in the de-canonicalizing dataset lines196-197, can you provide the performance of the proposed model for larger translation range?
-One of the concerns in capsule based architectures are scalability of these architectures, can you provide the classification and reconstruction performance on the larger subset of  ShapeNet?


**Time Spent Reviewing:**

10

---

> ### Author Response · Authors · 2021-08-06
> **Response to reviewer jD2y**
>
> Thank you for the positive feedback. Please see our responses below.
>
> ### Clarify the encoder architecture?
> We base our encoder on ACNe, which is a state-of-the-art architecture for processing point clouds. We modify this architecture to enable K-fold decomposition, specifically by extending its attention mechanism to allow for multiple attention. A detailed description is available in Section A.1 of the appendix.
>
> ### Does it only have one capsule layer with K capsule types as its last layer?
> Intermediate layers in ACNe **can** form K capsules, but we do not leverage this aspect. In each layer that utilizes Attentive Context Normalization (ACN), we could apply muti-headed ACN (line 558) to allow this to happen.
>
> ### Effect of K on the performance?
> This ablation study is available in Table 7 of the supplementary material.
>
> ### Considered adding “capsule layers” as intermediate layers?
> Our intermediate layers already perform K-fold attention, effectively acting as “capsule layers”.
>
> ### Provide performance of the proposed model for a larger translation range?
> Please see Reviewer 8HDQ’s reply “Experiments with a wide range of transformations”.
>
> ### Scalability of capsule architectures.
> Our performance is comparable to that of other point cloud encoder/decoder nets. Our work focuses on the extraction of **primary** capsules, whereas scalability becomes an issue in capsule architectures that discover capsule **hierarchies**. Hence we are not subject to this problem.
>
> ### Provide classification/reconstruction performance on a larger subset of ShapeNet?
> We are not sure we understand the request, could you please clarify?

---

> ### Comment · Reviewer_jD2y · 2021-09-11
> **Post Rebuttal**
>
> I appreciate the author'(s') clarification and the new experiments on a set of larger translation range.
> However, for the scalability issue, an experiment with larger number of classes (more that 13 classes) for the unsupervised classification task is necessary to support your claim and show case the quality of extracted features by primary capsules.
>
> Considering the rebuttal, I prefer to keep my score unchanged.

---

> ### Author Response · Authors · 2021-09-13
> **more than 13 classes?**
>
> We base our unsupervised training on an auto-encoding reconstruction task, which the community typically relies on the 13 class ShapeNet Core dataset since AtlasNet (2018+). While we would have loved to be able to launch our experiments on collections with more classes, this is what is currently publicly available / evaluated against, limiting our choices.
>
> We would highly appreciate any suggestions on a larger dataset that we could use to move beyond these 13 classes that we might have missed, or other practical ways that do not involve the extensive task of creating a new dataset, which in its own would be an enormous effort.

---

### Official Review · Reviewer_2DmA · 2021-07-16

**Rating:** 6
**Confidence:** 3

**Summary:**

The paper proposes a self-supervised capsule architecture for 3D point clouds. In order to circumvent the customary need to train on pre-aligned datasets, the authors introduce the Canonical Capsules to compute the K-part decomposition of a point cloud that allows unsupervised "object-centric" learning of 3D representations without requiring a semantically aligned dataset. The main idea of the canonical capsules is to enforce invariance/equivariance by relating the decomposition result of two random rigid transformations.  The authors compare the proposed method against state-of-the-art methods on 3D point cloud reconstruction, canonicalization, and unsupervised classification.

**Limitations And Societal Impact:**

Yes. The authors have addressed the limitation and potential negative societal impact of their work.

**Main Review:**

### Originality

The idea of the framework is basically enforcing the invariance/equivariance of the decomposition result of two random rigid transformations and use a canonicalizer to learn object-centric 3D representations. The idea is quite interesting and novel to some extent.


### Clarity

The paper is well written. The motivation and the idea of the proposed framework is well explained and could be followed even though I am not very familiar with this topic.


### Quality and Significance

I am not an expert on this topic. Overall, the method looks reasonable to me and the authors conduct comprehensive experiments.  One small question: could you please give more explanation why the proposed method becomes unstable when using RRI features?


**Time Spent Reviewing:**

7

---

> ### Author Response · Authors · 2021-08-06
> **Response to reviewer 2DmA**
>
> Thanks for the positive feedback. We address the comments below.
>
> ### Why unstable when using RRI features?
> We suspect this is due to the fact that RRI features discard information to achieve rotational invariance. This is somewhat analogous to how deep learning in images works better with raw images rather than post-processed ones (e.g. on SIFT descriptors).

---

### Official Review · Reviewer_WfrE · 2021-07-16

**Rating:** 8
**Confidence:** 4

**Summary:**

This paper proposes to learn a Canonical Capsule decomposition of shape point clouds in a self-supervised manner.
Given an unaligned 3D point cloud dataset, the proposed method constructs pairwise data with random SE(3) transformations, and then extracts the per-capsule pose and a transformation-invariant descriptor.
The experiment results show the importance of the object-centric representation with capsule decomposition in many applications, including reconstruction, canonicalization, and classification.


**Limitations And Societal Impact:**

Yes, the authors have adequately addressed the limitations and potential negative social impact of their work.

**Main Review:**

The proposed pipeline learns, in an unsupervised manner, the canonicalization of 3D point clouds from aligned/unaligned point cloud datasets. The method is described clearly and could be reproduced.
The comparison, in each application, against 3D-PointCapNet, AtlasNetV2, and AtlasNetV2 w/ STN, adequately shows the significant improvement brought by canonicalization.

I have a few questions:
1. When training on 13 categories, is there any kind of consistency between the corresponding capsules for point clouds coming from different categories?
2. It's better to show the reconstruction results of AtlasNetV2 w/ STN in Figure 2.

**Time Spent Reviewing:**

5 hrs

---

> ### Author Response · Authors · 2021-08-06
> **Response to reviewer WfrE**
>
> Thanks for the positive feedback. Please see responses below.
>
> ### Consistency between capsules from different categories?
> Note we did observe *some* consistency. For example, in the supplementary web page containing video examples, the “yellow” and the “blue” capsules are activated around the “endpoint” of pole-like objects (airplane’s head and tail, gun’s front and back). However, note we do not explicitly enforce this behavior among different categories, nor do we have the means to since there are no such labels in our data.
>
> ### AtlasNetv2 w/ STN in Fig2
> Thank you for the suggestion, we will add it in a revision.

---

### Official Review · Reviewer_8HDQ · 2021-07-16

**Rating:** 7
**Confidence:** 5

**Summary:**

The paper proposes a capsule-based network architecture that allows self-supervised representation learning from 3D point clouds.
While existing methods rely heavily on pre-canonicalized (aligned) 3D models for training, this work proposes to automatically discover the "optimal" canonical transformation for alignment while learning to decompose and reconstruct the input point clouds.
Specifically, it exploits the transformation invariance and equivariance properties of object parts (capsules) to learn a latent representation.
The experimental results on the (unaligned) ShapeNet dataset show that the method can effectively learn to decompose the 3D point cloud into consistent parts across objects and canonicalize unaligned data.
It outperforms the state-of-the-art methods in terms of reconstruction accuracy, canonicalization accuracy and stability, as well as unsupervised classification accuracy from latent representation.
It is very encouraging to see that the need of 3D pose/viewpoint annotations can be removed and such competitive performance is reached.

**Limitations And Societal Impact:**

The authors have adequately addressed the limitations and potential negative societal impact of their work in Section 5.

**Main Review:**

[Motivation and problem setting] \
The paper addresses the problem of 3D representation learning from point cloud data, where the 3D point clouds are not aligned in the canonical space.
Compared to most existing works that assumes the availability of aligned 3D data, the proposed scenario is more practical and crucial since such pose/viewpoint annotations of real-world 3D objects are usually hard to obtain.
By removing the need of pre-canonicalized 3D data for training, I believe this work makes a significant step towards learning from real-world 3D data with diverse transformations.


\
[Method] \
The network design, training strategies, and losses are well-justified and technically sound.
The proposed method follows an interesting line of work with part-based representation.
Object parts are shown to be a good mid-level representation which is faithful to structural details, consistent across instances, and beneficial for pose/shape ambiguity.

From my understanding, equations (5) and (10) are individually solved in each branch in Figure 1 with distinct transformation ($T^a$ or $T^b$).
That is, $\theta_k$ and $\bar{\theta_k}$ are both from the top branch (${\theta_k}^a$ and ${\bar{\theta_k}}^a$) or both from the bottom branch  (${\theta_k}^b$ and ${\bar{\theta_k}}^b$).
If so, Figure 1 is a little confusing since $\theta_k$ and $\bar{\theta_k}$ for shape matching seem to come from the top and bottom branch, respectively.


\
[Experiments] \
The experimental settings with de-canonicalized (unaligned) ShapeNet data and mStd metric are reasonable.
The random transformation includes rotation and translation within the range [ 0.2, 0.2].
I am curious if the authors have experimented with 1) a wider range of translation, 2) random scaling, and 3) decoupling the three transformations (e.g. only apply rotation).
Since addressing unaligned training data is a major contribution of this paper, it would be important investigate the effect of different transformations.

Both the quantitative and qualitative results show a significant and consistent improvement compared to the state-of-the-art methods on the synthetic ShapeNet dataset.
It is not necessary but would be better to compare with more existing methods and on multiple datasets.

For auto-encoding reconstruction, is it possible to also evaluate the performance of different part-based representations like multi-mesh and implicit functions?
It should be possible to sample 3D points from mesh or implicit functions and calculate Chamfer distance.
I believe this would make the proposed representation more convincing, and the results on unaligned data would be very helpful for follow-up research.

Table 7 in the supplemental document shows the ablation studies on the number of capsules.
Are the results from multi-class training and evaluation?
It seems that there is an optimal number of capsules to best represent the 3D objects, and I suspect it varies between object classes.
I am also curious if the authors have tried fixing the descriptor dimension while increasing the number of capsule, since it is strange that more capsules don't lead to higher reconstruction accuracy.

I find it surprising that the network can figure out the optimal canonical pose and capsule configuration of multiple classes as the same time.
The detailed (per-class) quantitative results of multi-class training are missing, and the paper does not specified whether the qualitatively results are from single-class or multi-class training.
It is hard to observe the pros and cons of multi-class training.

Finally, I wonder if the authors have observed any failure cases of the proposed method.
I would like to see more discussion about the limitation of representation and more qualitative results of failure cases.


\
[Writing] \
The paper is well-written and easy to follow.

It is a little unclear to me which parts (losses/networks/representation) of the proposed method are from the baselines (AtlasNetV2 and 3D-PointCapsNet) and which are novel.
Maybe the authors can clarify it in the introduction or related work.

The word "object-centric" is mentioned multiple times.
Do "object-centric reference frame" in L116 and "object-centric coordinate frame" in L329 both mean the (optimal) canonical frame found by the network?


\
[Reference] \
Some missing reference I feel relative to the topic:
[1] Luo, Tiange, et al. "Learning to Group: A Bottom-Up Framework for 3D Part Discovery in Unseen Categories." ICLR. 2019.
[2] Paschalidou, Despoina, Luc Van Gool, and Andreas Geiger. "Learning unsupervised hierarchical part decomposition of 3d objects from a single rgb image." CVPR. 2020.
[3] Kawana, Yuki, Yusuke Mukuta, and Tatsuya Harada. "Neural star domain as primitive representation." NeurIPS. 2020.


**Time Spent Reviewing:**

8

---

> ### Author Response · Authors · 2021-08-06
> **Response to reviewer 8HDQ**
>
> We appreciate the thoughtful and positive feedback. Please see our comments below.
>
> ### Figure 1 is a little confusing
> Your understanding is correct. We will clarify the figure to make this clearer.
>
> ### Experiments with a wider range of transformations
> - Note that large translations can be easily reduced to small ones by centering around the cloud's center of mass. We just launched an experiment to validate this, and hope results will be back in time to communicate this back.
> - With respect to scale, note that while this quantity can also be estimated by Procrustes/SVD, typical point clouds are metrically scaled, as they are acquired by, e.g., lidar sensors (i.e. scale regression is not necessary).
>
> ### Different representations such as multi-mesh and implicit functions?
> Applying other representations for the decoder should indeed be a straightforward extension of our method. While in this paper we focus on the introduction of the framework, a study on how the framework performs with other representations is left for future research.
>
> ### Ablation study on the number of capsules (Table 7)
> Below, we answered this question in three parts:
>
> #### Single-class or multi-class training for this experiment
> All ablation studies (in both main paper and supplementary) are executed on the airplane class (the shapenet class with the largest cardinality), so to allow us to conduct experiments more efficiently, and leverage our compute budget to ablate with respect to more variations. Using multi-class training for the selection of the number of capsules would likely result in a boost in performance of our method, benefitting from improved hyperparameters.
>
> #### Consistent descriptor dimension
> Keeping the descriptor dimension the same increases the reconstruction performance, since the capacity of the network grows linearly w.r.t. this parameter (we launched an experiment in regards, and will report results if/when ready). However, in this experiment we wanted to keep the capacity of the network the **same**, and understand whether using “more/weak” capsules was better than using “fewer/strong” capsules.
>
> #### Why the larger number of capsules doesn’t help
> Our results reveal that using more capsules that are with lower-dimensional descriptors can cause performance degradation. This is unsurprising, as the descriptors become too small to capture the geometric variations within the same capsule (recall the overall network **capacity is kept fixed**).
>
> ### Are qualitatives single or multi-class?
> Qualitative results in the main paper are from multi-class training. Note in the supplementary web page we visualize both multi as well as single class results.
>
> ### Wonder if authors have observed any failure cases
> The most common type of failure case arose when the hyper-parameters of the self-supervised losses were not tuned properly, leading to mode collapse in the capsule decomposition.
> Another failure case is that objects within the **same** category with very different geometry (e.g. couch vs. stool) might not be consistently aligned (e.g. “up” direction might differ).
>
> ### Which parts are from prior works.
> We adapt the encoder and the decoder architectures from ACNe and AtlasNetV2 for our purpose. These are not the original architectures, but were modified to allow for multi-head outputs (ACNe) or per-capsule latent rather than a global latent (AtlasNetV2). Details can be found in Supplementary/PartA.
>
> ### “Object-centric reference frame” vs. “Object-centric coordinate frame”
> Yes, the two terms refer to the same concept, we will consolidate our text in a revision.
>
> ### References
> Thanks, we will incorporate the suggested references in a revision.
>
> ### Per-class quantitative are missing
> We will add the table below to our supplementary (metric is CD as in Table 1).
>
> #### Aligned case
> |            | 3D-PointCapsNet | AtlasNetV2 | Our method |
> |------------|-----------------|------------|------------|
> | Bench      | 2.06            | 1.67       | **1.44**   |
> | Cabinet    | 3.23            | 2.81       | **2.37**   |
> | Car        | 2.64            | 2.42       | **2.10**   |
> | Cellphone  | 2.25            | 2.00       | **1.77**   |
> | Chair      | 2.64            | 2.26       | **1.90**   |
> | Couch      | 2.99            | 2.63       | **2.26**   |
> | Firearm    | 0.93            | 0.78       | **0.59**   |
> | Lamp       | 3.40            | 2.68       | **1.61**   |
> | Monitor    | 2.85            | 2.52       | **2.09**   |
> | Airplane   | 1.36            | 1.18       | **0.99**   |
> | Speaker    | 4.26            | 3.80       | **3.04**   |
> | Table      | 2.56            | 2.16       | **1.80**   |
> | Watercraft | 2.05            | 1.73       | **1.31**   |
> | All        | 2.49            | 2.14       | **1.76**   |
>
> #### Unaligned case
> |            | 3D-PointCapsNet | AtlasNetV2 | AtlasNetV2 w/ STN | Our method |
> |------------|-----------------|------------|-------------------|------------|
> | Bench      | 4.34            | 2.93       | 2.44              | **1.84**   |
> | Cabinet    | 5.20            | 3.65       | 3.30              | **2.72**   |
> | Car        | 4.57            | 3.46       | 3.10              | **2.45**   |
> | Cellphone  | 3.87            | 2.46       | 2.17              | **1.86**   |
> | Chair      | 5.55            | 3.53       | 2.93              | **2.72**   |
> | Couch      | 5.33            | 3.72       | 3.28              | **2.70**   |
> | Firearm    | 2.00            | 1.28       | 0.96              | **0.67**   |
> | Lamp       | 4.97            | 2.91       | 2.56              | **2.20**   |
> | Monitor    | 4.30            | 3.06       | 2.70              | **2.46**   |
> | Airplane   | 3.24            | 2.17       | 1.60              | **1.30**   |
> | Speaker    | 6.05            | 4.42       | 3.96              | **3.55**   |
> | Table      | 5.17            | 3.20       | 2.67              | **2.36**   |
> | Watercraft | 3.38            | 2.27       | 1.86              | **1.53**   |
> | All        | 4.66            | 3.08       | 2.60              | **2.22**   |

---

> > ### Author Response · Authors · 2021-08-10
> > **Response to reviewer 8HDQ**
> >
> > As promised, we report the results of additional experiments below.
> > In summary, the experimental results are consistent with our review responses.
> >
> > ### Larger range of translation
> > We increase the range of random translations ([-0.2, 0.2] in the original paper). As shown in the table below, we observe the negligible changes in reconstruction performance (CD) with larger random translations.
> >
> > | Range of translation | **[-0.2, 0.2]** | [-0.4, 0.4] | [-0.8, 0.8] | [-1.6, 1.6] |
> > |----------------------|-------------|-------------|-------------|-------------|
> > | CD                   | 1.11        | 1.12        | 1.10        | 1.10        |
> >
> >
> > ### Fixing the descriptor dimension while increasing the number of capsules
> > Note this is different from Table 7, as in that setting the overall network capacity was kept fixed.
> > As shown in the table below, perhaps unsurprisingly, more capsules lead to better reconstruction performance
> >
> > | Number of capsules | 5    | 10   | 20       |
> > |--------------------|------|------|----------|
> > | CD                 | 1.51 | 1.11 | **1.02** |

---

### Official Review · Reviewer_WpqU · 2021-07-19

**Rating:** 7
**Confidence:** 5

**Summary:**

This paper proposes a way of learning a capsule-based representation of 3D point clouds. The representation decomposes an input point cloud into parts, each corresponding to a capsule. This decomposition is trained to be consistent across SE(3) transformations of the point cloud using loss terms that encourage feature invariance and pose equivariance across two random presentations of the input point cloud. Moreover, the model learns a canonical pose for each part that is tied to the identity of the object (not its presented pose).

The experiments show that
1. The reconstruction quality is high, showing that the model has capacity to fit point clouds from ShapeNet.
1. The canonical pose is indeed consistent (measured by low std deviation in relative angle of rotation).
1. The model can be used for pairwise registration (but DeepGMR does even better using RRI features, but doesn't do any of the other things this model can do.)
1. The learned features lend themselves well to unsupervised clustering/classification.
1. All the proposed loss terms are useful and contribute to better reconstruction and canonicalization.

**Limitations And Societal Impact:**

Yes

**Main Review:**

Strengths
1. The model is able to find a part decomposition that is consistent across presented views, while retaining the ability to autoencode and classify.
1. The stability of the canonical pose is impressive.
1. Good performance on standard tasks and benchmarks.
1. Adequate comparisons to previous work.


Weakness
1. Capsules are not interchangeable. For instance, if I understood correctly, given an airplane point cloud, the encoder always assigns the 1st capsule to say, the left wing, 2nd capsule to the nose, etc. In other words, the capsule indices are tied to the entity they represent, as opposed to having an unordered *set* of capsules which can be constituted into different objects. This limits the ability of this model to reuse capsules across different object types and deal with the presence of multiple objects in the same scene.
1. The model has no way to "turn off" capsules, which means that when trained on multi-category data it is forced to use the same K capsules very differently depending on the object being encoded. It is surprising however that the model still works very well for pair-wise registration and classification. It would be useful to include a brief discussion on whether the presence probabilities can be modeled within the framework of this model (similar to the Stacked Capsule Autoencoders paper).
1. The "pose" of the part is really just the centroid. It does not capture the orientation of the part, which is something one typically associates with pose. It would be useful to include a comment on why orientation was not used (perhaps it is not helpful ?).
1. Removing canonicalization loss does not seem to have a big impact (Table 4). This is perhaps explained by the fact that the reconstruction loss already encourages canonicalization because it minimizes the CD between the canonicalized input point cloud and the reconstructed one ? It would help to clarify this.

Overall, this paper makes a significant contribution by proposing a model that learns canonical part decompositions in canonical pose, while retaining the ability to remember object shape and identity.

**Time Spent Reviewing:**

5

---

> ### Author Response · Authors · 2021-08-06
> **Response to reviewer Wpqu**
>
> Thanks for the insightful and positive feedback. Please see our detailed response below.
>
> ### Capsules are not interchangeable (the ordered capsules have some disadvantages).
> Having interchangeable capsules is one of the necessary conditions in allowing our method to be applicable to scenes with multiple objects (i.e. instance segmentation), a future direction we note in line 337. We are actively investigating how this could be achieved in follow-up work.
>
> ### The model has no way to turn off capsules
> This is indeed the case, and relevant to allowing our method to be applicable to incomplete point clouds (line 337). We are investigating this aspect, but the results are too preliminary to be shared at this point (it seems to work but more thorough experiments are needed), and are beyond the scope of this submission.
>
> ### It does not compute the orientation of parts
> We did attempt explicit rotational representations early-on in our experiments, but the performance was not as satisfactory, likely due to the difficulty in representing rotational quantities within deep networks is well known, and an active topic of research (Levinson et al. An analysis of SVD for deep rotation estimation. NeurIPS 2020).
>
> ### Removing canonicalization loss does not seem to have a big impact
> Indeed, in **Table 10 of the supplementary material**, we show that:
> - If the task is to canonicalize, then reconstruction loss can be omitted
> - If the task is to reconstruct, the canonical loss leads to slightly better CD/mStd
>
> The full model is preferable when one desires the best performance, but if the objective is **only** canonicalization, then the additional compute resources necessary to train the decoder parameters, and to compute the O(P log P) distances (where P is the number of points in the point cloud) required for CD computation can be avoided.

---

### Decision · Program_Chairs · 2021-09-27

**Decision:**

Accept (Poster)

**Comment:**

The authors have successfully addressed the raised issues. All reviewers agree that this paper makes good contribution to the NeurIPS community. The area chairs agree and recommend this paper be accepted.